# Antimicrobial Peptides as an Alternative for the Eradication of Bacterial Biofilms of Multi-Drug Resistant Bacteria

**DOI:** 10.3390/pharmaceutics14030642

**Published:** 2022-03-15

**Authors:** Janaína Teixeira Costa de Pontes, Anna Beatriz Toledo Borges, Cesar Augusto Roque-Borda, Fernando Rogério Pavan

**Affiliations:** 1Tuberculosis Research Laboratory, School of Pharmaceutical Science, São Paulo State University (UNESP), Araraquara 14800-903, Brazil; j.pontes@unesp.br (J.T.C.d.P.); anna.beatriz@unesp.br (A.B.T.B.); cesar.roque@unesp.br (C.A.R.-B.); 2Vicerrectorado de Investigación, Universidad Católica de Santa María, Arequipa 04013, Peru

**Keywords:** antimicrobial peptides, antimicrobial resistance, biofilm, drug discovery, extensive drug resistance, MDR bacteria

## Abstract

Bacterial resistance is an emergency public health problem worldwide, compounded by the ability of bacteria to form biofilms, mainly in seriously ill hospitalized patients. The World Health Organization has published a list of priority bacteria that should be studied and, in turn, has encouraged the development of new drugs. Herein, we explain the importance of studying new molecules such as antimicrobial peptides (AMPs) with potential against multi-drug resistant (MDR) and extensively drug-resistant (XDR) bacteria and focus on the inhibition of biofilm formation. This review describes the main causes of antimicrobial resistance and biofilm formation, as well as the main and potential AMP applications against these bacteria. Our results suggest that the new biomacromolecules to be discovered and studied should focus on this group of dangerous and highly infectious bacteria. Alternative molecules such as AMPs could contribute to eradicating biofilm proliferation by MDR/XDR bacteria; this is a challenging undertaking with promising prospects.

## 1. Introduction

Bacterial resistance is a current emergency problem that has claimed millions of lives in recent years [1]. The hospital environment is the main place in which these deaths occur and, normally, the cause of this lethal bacterial resistance is the non-elimination of all petrogenic microorganisms in the treatment of patients. Thus, the microorganisms remaining after this insufficient treatment proliferate despite the use of antibiotics [2]. The recurrence of cases such as this leads to the resistance of several bacteria to different drugs, allowing their massive proliferation, which, compounded with the ability to form biofilm, makes them more resistant and more difficult to fight [3]. Bacterial resistance can be innate or acquired since many bacteria have resistance genes that are expressed only when they feel threatened; and they can easily transfer these genes when found in microecosystems to other species for community survival [4].

Naturally, bacteria capable of forming biofilms are practically everywhere. To form biofilms, these microorganisms adhere to surfaces, which can be biotic (composed of living beings or parts of them) or abiotic (composed of non-living substances). The biofilms cover the surfaces with an extracellular polymeric substance (EPS) [3]. This substance has conglomerates of proteins, polysaccharides, and exogenous DNA [5]. Diseases related to these bacteria develop slowly, allowing for greater adaptation and production of biofilm and subsequently leading to severe local inflammation [6]. The increasing cases of hospital infections, mainly highly lethal urinary tract and bloodstream infections caused by biofilm-forming bacteria in medical equipment, have concerned the World Health Organization (WHO) and led it to publish a list of priority bacteria for study and drug discovery [7,8,9].

Herein, we summarize the resistance mechanisms that can occur in bacteria. We focus on the formation of biofilm and the general characteristics and particularities of each source of biofilm, based on some species from the list of critical and high-priority bacteria. We also present some control mechanisms that are known today and concentrate on antimicrobial peptides (AMPs) as an alternative in the fight against biofilm formation by some species of multi-drug resistant (MDR) or extensively drug-resistant (XDR) bacteria [4]. To our knowledge, this is the first article reporting the importance of AMPs against biofilm-forming bacteria on the critical and high WHO priority list.

### 1.1. Antimicrobial Resistance

Antimicrobial resistance (AMR) could result in a setback for modern medicine due to the difficulty in the control of a series of infectious pathologies, caused by the few options available for treatment against resistant strains. This situation has led AMR to be considered one of the biggest challenges to health in the 21st century. According to the United Nations Interagency Coordinating Group on AMR, microbial resistance is linked to 700,000 deaths every year, and it is believed that by the year 2050 it will be responsible for 10 million deaths [10]. The inefficiency of the standard treatment has resulted in the persistence of bacterial infections [11], associated with both the development of resistant strains and the inadequacy of antibacterial treatment, the latter being attributed to its low spectrum of action or its high toxicity [12].

The impaired quality of life, combined with the development of systemic bacterial infections, the increase in recurrence rates, chronicity, and the development of future opportunistic infections with resistant organisms are the consequences of bacterial infections caused by antibiotic-resistant organisms, such as MDR, XDR, and pan-drug-resistant (PDR) strains [13]. Therefore, AMR is a serious problem that affects several areas beyond modern medicine and the pharmaceutical industry, intensified by the resulting serious socio-economic and ecological impacts [7].

AMR is also related to the low development of new molecules. The pharmaceutical industry fails to invest in the development of new antibiotics because of the high cost and time required to successfully bring antimicrobials to the market. The fact that the molecule may prove inefficient shortly after its introduction, the increase in regulatory conditions, and the strict price controls imposed by many governments are some of the most influential reasons discouraging investment in the development of new antibiotics [11].

The emergence and spread of resistance mechanisms among bacteria is a consequence of the small investment and long time required for the search and development of antibiotics, as well as the fact that the development of long-term-usage drugs is economically more attractive. Such circumstances have deteriorated public health worldwide, as shown by the increasing need of hospital treatment for bacterial infections that require higher doses and a longer hospital stay [2].

The administration of antimicrobials can cause bacteriostatic, bactericidal, or bacteriolytic effects. The bacteriostatic effect results from the use of medication that stalls bacterial cellular activity without directly causing bacterial death [14]. The bactericidal effect occurs when the drug results in bacterial death. The bacteriolytic effect takes place when an antimicrobial causes the dissolution and destruction of the bacteria.

Bacteria are one of the oldest life forms on Earth, and they have been around for billions of years. They have acquired a strong adaptive ability due to their genomic flexibility in shielding themselves from toxic chemicals. In addition, due to their capacity of upkeep and transfer of genes, bacteria have been responsible for AMR and the dissemination of antimicrobial-resistant genes through an exchange of genetic material between different species [2,15].

AMR occurs when there is a fragile balance between bacteria and drugs, giving rise to selective pressure of bacteria prone to developing resistance; this ends up surpassing the relationship between new bacterial resistance/drug discovery, since in recent years the number of drugs approved by the FDA has decreased [16]. This collective manifestation of selective pressure, as well as its relationship with bacterial ecology, involves genetic material grouped in introns, which can, for example, provide an entire ecological system with resistant survival and tolerance traits [17].

### 1.2. AMR Mechanisms

Antibiotics are known to have a variability of targets for causing death or preventing bacterial proliferation. These targets can be the cell wall or cell membrane, inhibition of essential protein, or nucleic acid synthesis, among others. AMR mechanisms involve “running away” from the action of each antibiotic caused by early treatment abandonment, dose-interval error, missed dose, or improper sharing of antimicrobials [2]. Nowadays, there are bacterial strains resistant to several types of antibiotics, the so-called “multi-resistant forms”, which have become a great public health problem in several countries [18].

One of the AMR mechanisms is intrinsic resistance, i.e., the natural resistance of some bacteria to some antibiotics to which they have never been exposed but are insensitive to because of their inner cellular defense mechanisms [19]. Another AMR mechanism is mutation, which naturally occurs in every cell proliferation, but an increase in the number of bacterial cells with resistance-conferring mutation occurs due to selective pressure in the environment, which can be generated precisely by the antibiotic along with its misuse [7].

Antibiotics may also be inactivated through the production of enzymes capable of preventing the drug from reaching its target in the microorganism. For example, antibiotics that have the β-lactam structure may be inactivated by enzymes called beta-lactamases, which are capable of cleaving the beta-lactam ring and inactivating the drug [20]. This type of resistance includes resistance to penicillins, carbapenems, cephalosporins, and monobactams [7]. In the case of AMR to the quinolone class (whose mechanisms of action target essential bacterial enzymes, namely DNA gyrase and DNA topoisomerase IV), there are three resistance mechanisms: mutations that alter drug targets (protecting DNA enzymes), mutations that reduce drug accumulation by increasing the expression of efflux pumps, and the presence of plasmids that protect bacterial cells from the lethal effects of quinolones [21].

Polymyxin antibiotics are based on their mechanism of action, i.e., the formation of pores in bacterial cell membranes, causing leakage of the internal contents and the death of the bacteria. The use of this class of antibiotics has increased as a last-line therapeutic option against several MDR bacteria, which contributes to the emergence of AMR. In the case of polymyxins, AMR may be related to the loss of a group with a negative charge on the bacterial membrane, reducing its affinity with the antibiotic, which, in turn, is a cationic molecule; alternatively, it may be related to increased rigidity of the cell membrane in defensive response to the environment [22].

The AMR can also occur by horizontal gene transfer (a kind of gene transfer that occurs between different bacterial species), which can take place by “transformation” (the uptake of exogenous DNA by the bacteria), “conjugation” (when a bacterium transfers a conjugative plasmid to another through a structure called “conjugative pilus”), or by “transduction” (a process that includes the integration and passive replication of a viral genome into bacterial chromosome through the action of bacteriophages) [18]. Finally, other AMR mechanisms involve efflux pumps (transmembrane proteins capable of blocking drug entry or extrusion) and biofilm formation (by increasing its density and quorum sensing and consequent inhibition of antibiotic penetration through the biofilm matrix) [21].

### 1.3. Biofilms

Biofilms are communities of microorganisms that cooperate in an organized system associated with an appropriate substrate, in a way that resembles social cooperation (Figure 1); this phenomenon is the result of the evolutionary capacity that some microorganisms have acquired to protect themselves from the environmental threats that a solitary microorganism is not able to tolerate [23]. Biofilms contain the microorganisms themselves embedded in a self-produced extracellular matrix or EPS, made up of a series of substances that provide it with nutrients and basic survival resources; in addition, the cell–cell and cell–substrate adhesive capacity forms a 3D structure with different architectures and chemical compositions. EPS can contain substances produced by the same microorganisms, in general, organic compounds, mineral salts and water; or substances that are obtained from the infected host, such as excreted fluids, serum, and saliva derivatives.

Bacterial biofilms have different characteristics in terms of growth capacity, gene expression, and protein production. In addition, their resistance mechanisms are different from those of solitary bacterial cells, such as their ability to tolerate a certain level of environmental stress [24]. EPS may vary according to the species producing the system and/or according to variations in the environment, maintaining a nutritionally rich protection system, which contributes to proliferation, communication via signaling by quorum sensing, exchange of molecules, and horizontal transfer of genetic material, i.e., a process whereby a transfer of genes occurs between non-descendant cells. Obviously, for the treatment against this phenomenon, conventional drugs must be used in doses high enough to penetrate the system and reach the bacteria [23].

### 1.4. Control Mechanism

Given that biofilms increase bacterial pathogenicity and resistance to antibiotics, there is a need to formulate a mechanism aimed at controlling biofilm growth, since in many bacteria there is no way to eliminate an entire biofilm, only to reduce it [25]. Such a pathogenic system, in some cases, has the ability to protect invading bacteria against the host’s immune system through altered activation of phagocytes and the complement system, generating a less accurate immune response [6].

One of the control strategies is the use of antibiotics, but due to their toxicity to the human body in general and side effects, it is not possible to reach the minimum inhibitory concentration (MIC) of antibiotic in vivo. In addition, the ability to form biofilms can increase the AMR by 1000 times, which makes it very difficult to fight bacterial infections involving biofilms with only one antibiotic [26].

Wu et al. [6] reported that one of the main causes of the increase in biofilm infections is the use of foreign bodies and that the treatment for this is the removal of the infected object, replacement with a new uninfected one, and aggressive administration of antibiotics. A combination of antibiotics is usually previously selected based on the sensitivity of the microorganism in question and the ability of the drugs to adequately penetrate the infecting biofilm matrix [6,27].

Some strategies consist of using a hydrophilic polymeric coating in the construction of antifouling surfaces that reduce microbial adhesion, a coating that can be combined with antibiotics or disinfectants [28]. Other treatments are photodynamic therapy, involving the use of a photoactive dye and subsequent irradiation in the presence of oxygen, which has a bactericidal effect, as well as the use of antibiofilm molecules, already proven to be effective, and biofilm-dissolving substances. It is known that most antibiofilm molecules interfere with bacterial signaling pathways (both in Gram-positive and Gram-negative bacteria), and they can be enzymes, peptides, antibiotics, polyphenols, among others [25]. A bacteriophage cocktail is a way of using anti-biofilm molecules. This treatment is responsible for delaying the appearance of phage-resistant bacteria because it targets different host receptors [29]. Phages with different ranges of lytic activity are used in order to increase their lytic activity, extend the phage host range, and increase the number of target pathogens, and if combined with antibiotics, the phages improve their anti-biofilm properties [30,31].

Many anti-biofilm substances have been identified, most of which were isolated from a natural force and some consist of synthetic compounds, chelating agents, and antibiotics [25]. Therefore, it can be concluded that there is a great diversity of anti-biofilm mechanisms since several anti-biofilm compounds exist. Some examples of such mechanisms are as follows.

(i) Inhibition of quorum sensing: the use of signaling molecules for colony communication, population control, and swarm motility, through inhibition of the LuxR-type transcriptional activator protein. This protein regulates the expression of the target gene responsible for the LuxI-like synthase protein, which synthesizes these signaling molecules [32]. (ii) Peptidoglycan cleavage: a method based on tannic acid, a polyphenolic compound that inhibits biofilm formation in *Staphylococcus aureus*, without affecting bacterial growth [33]. This method is capable of reducing biofilm formation in several ways, such as the release of signaling molecules related to biofilm gene expression into the extracellular environment, altering the composition of proteins and teichoic acids that form the cell wall.

The main biofilm-forming bacteria listed by WHO-list priority as *Pseudomonas aeruginosa*, *Acinetobacter baumannii*, *Enterococcus* sp., *Staphylococcus aureus*, *Klebsiella pneumoniae* and *Mycobacterium tuberculosis* are described in Table 1.

## 2. Antimicrobial Peptides and Applications

AMPs are biomolecules formed by amino acids that vary in length, usually composed of 12–50 amino acids [60]. They are known to have great antifungal, antiviral, and antibacterial properties and are capable of reducing the bacterial load and avoiding resistance due to their ability to associate rapidly with the membrane [7]. AMPs are also small protein fractions with biological activity and are part of the body’s first line of defense for pathogen inactivation [61]. The first AMPs discovered and studied were based on structures related to defensins since these molecules were produced innately when some pathogenic agent came into contact with organisms [4]. AMPs are capable of modulating the immune system and generating a better response to defend the host since previous studies would indicate this potential as a single molecule [62,63,64], or in combination with another drugs, causing an even more beneficial and less toxic synergistic effect [65,66].

AMPs have an amphipathic nature because they are composed of hydrophilic and hydrophobic regions, although they are mostly hydrophobic. This allows them to interact with biological membranes due to van der Waals interactions with the membrane lipid tails, which are natural in cell membranes [67]. Most AMPs have a cationic behavior that promotes the interaction with membrane headgroup components [68,69]. They can adopt different secondary structures that influence their mechanism according to their physicochemical characteristics (Figure 1); the mechanism of action is also influenced by the net charge, amphipathicity, and number of amino acids [70]. AMPs act against bacteria due to membrane disruption and/or pore formations. Other actions consist of the inhibition of proteins, enzymes, and cell wall synthesis when they are present in the cytoplasm [67]. Due to this ability, shown in many studies, AMPs are considered effective against MDR bacteria and fungi cells [71].

The neutralization or disassembly of lipopolysaccharides in these strategies uses AMPs, which can penetrate through the lipid bilayer, since they have a hydrophobic side and a hydrophilic side, allowing their solubilization in an aquatic environment [63]. AMPs are able to infiltrate the biofilm and cause bacterial death [72] due to their ability to electrostatically bind to lipopolysaccharides (LPS), involving interaction between two cationic amino acids (lysine and arginine) and their respective heads of groups, forming a complex. This complex destabilizes lipid groups due to the formation of multiple pores, impairing the integrity of the bacterial cell membrane [73]. This is due to the fact that the complex is stabilized through hydrophobic interactions between the hydrophobic amino acids of the peptide and the fatty acyl chains of LPS [63].

AMPs are considered an excellent alternative against resistant bacteria, in comparison with conventional antibiotics. This is due to their non-specific mechanism (ability to reach a variety of sites), which reduces the chances of resistance development. In addition, some AMPs have great anti-multi-resistant biofilm activity, interfering with the initiation of biofilm formation (preventing bacteria from adhering to surfaces) or destroying mature biofilms (killing the bacteria present or causing them to detach) [74]. Table 2 presents some examples of peptides with activity against bacteria and their properties.

Studies have shown that AMPs are effective in degrading bacterial biofilm, although the mechanism is poorly known. According to Yasir et al. [74], the five main antibiofilm mechanisms are as follows: interruption of quorum sensing; that is, of the bacterial cell signaling systems of the biofilm; disruption or degradation of the membrane potential of cells belonging to the biofilm; alarm system inhibition, preventing strict bacterial response; degradation of the polysaccharide and biofilm matrix; dysregulation of genes responsible for biofilm formation and transport of binding proteins. However, the mechanism of biofilm resistance has been associated with bacterial adaptation, heterogeneity of the bacterial cells, combined use with antibiotics, and interaction with EPS. The last mechanism is caused by the fact that most AMPs possess a positive charge. When EPS possesses a negative charge, it allows it to trap the AMPs, inhibiting their actions against the bacteria [71].

According to Wang et al. [99], AMPs may have the ability to inhibit the expansion of biofilms, and not always eliminating all microorganisms such as Nal-P-113 against *Porphyromonas gingivalis* W83 biofilms formation; therefore, the authors suggest its application with other drugs currently used for the oral treatment of this potentially virulent bacterium. Likewise, some studies report that their synergistic or combined effect could improve with the inclusion or structural modification of AMP; for example, chimeric peptide-Titanium conjugate (TiBP1-spacer-AMP y TiBP2-spacer-AMP) against *Streptococcus mutans*, *Staphylococcus epidermidis*, and *Escherichia coli* [100], A3-APO (proline-rich AMP) combined with imipenem against ESKAPE pathogens, biofilm-forming bacteria and in vivo murine model [101,102]. In addition, it was reported that modifications in the C terminal with fatty acids could further improve the specificity and activity of AMPs against superbugs and their respective biofilms [103,104]. Another study revealed that the addition of a hydrazide and using perfluoroaromatic (tetrafluorobenzene and octofluorobiphenyl) linkers enhance the antibacterial and antibiofilm activity demonstrated against MDR and XDR *A. baumannii* [105]. Table 3 shows promising applications of AMPs against biofilm formation.

## 3. Perspectives and Conclusions

To conclude, there is great urgency in terms of public health for the development of new treatments against multi-drug resistant and extensively antibiotic-resistant bacteria. The main bacteria of concern are well known as they are listed by the World Health Organization. In addition, this review describes the main bacteria causing biofilm formation. We noted that there are several mechanisms of bacterial resistance, and that the formation of biofilm is one of the most worrisome. Additionally, each biofilm-forming bacterial species forms this complex structure with different particularities at structural and molecular levels and, therefore, each species needs different treatments.

We show that many AMPs have excellent antimicrobial activity, but that they can also be potential inhibitors of biofilms. We also show that AMPs can serve to potentiate obsolete conventional antibiotics, to generate a synergistic effect to eradicate bacteria as well as their respective biofilms. Nanotechnology is another important tool during the eradication of biofilms and MDR/XDR bacteria, increasing the specificity, controlled release of the drug/peptide, decreasing toxicity, and increasing its bioavailability [134]. In addition, bioconjugation also demonstrated significant results against MDR and XDR biofilm-forming bacteria [105]. We conclude that studies on drug discovery using AMPs are promising and that these peptides can be an alternative in the fight against infections by MDR-bacterial infections and biofilm formation. Finally, we encourage and emphasize further studies involving biofilm-forming bacteria included on the WHO priority list, as these bacteria are becoming more dangerous every day.

## Figures and Tables

**Figure 1 pharmaceutics-14-00642-f001:**
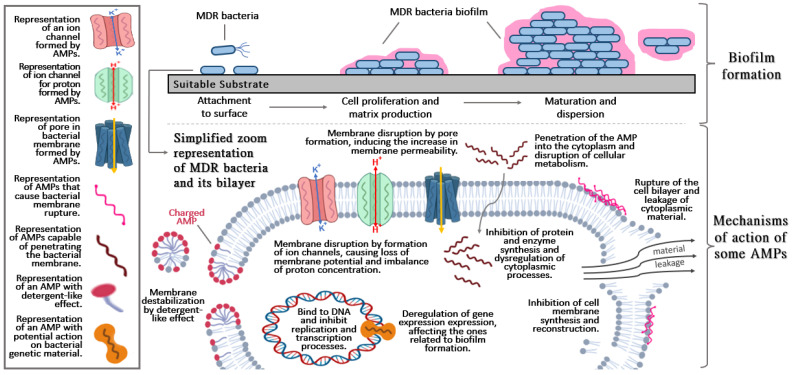
Basic and general aspects of biofilm formation (**upper section**) and brief description of the mechanisms action of antimicrobial peptides (AMPs) (**lower section**).

**Table 1 pharmaceutics-14-00642-t001:** The main biofilm-forming bacteria listed by WHO-list priority.

Bacteria	Mechanism of Biofilm Formation
*Pseudomonas aeruginosa*	*Pseudomonas aeruginosa* is one of the main opportunistic pathogens due to its high virulence, causing great concern in terms of public health. This microorganism is associated with several serious infections such as ventilator-associated pneumonia and chronic infections in cases of cystic fibrosis. *P. aeruginosa* has a pathogenic profile that makes it capable of developing resistance to several antibiotics. This is due to its complex genome, its various virulence factors, and its ability to form powerful biofilms [34]. *P. aeruginosa* forms a dense biofilm as their microcolonies start forming stalks that eventually develop into “mushroom” structures. This bacterial species requires polysaccharides in the extracellular matrix of its biofilm at most stages of maturation. Two classes of polysaccharides are important in biofilm formation: capsular polysaccharides, which maintain the outer structure of cells with dynamic protective polymers, and aggregative polysaccharides, which offer structural integrity and interaction with other matrix components, conferring an adhesive and aggregative capacity to the biofilm, mainly by the alginate produced by *P. aeruginosa* [35].
*Acinetobacter baumannii*	*Acinetobacter baumannii* is a Gram-negative opportunistic nosocomial pathogen, known to have multi-resistance to various antibiotics due to, among other factors, its major virulence, which is its ability to produce biofilms [36]. This bacterium has the ability to survive and proliferate in favorable and unfavorable environments, such as hospital equipment. This ability to colonize abiotic environments (such as polystyrene and glass) is believed to be associated with its power to form characteristic biofilms [37]. An important organelle responsible for the initial fixation of this microorganism to surfaces is the pilus, which allows bacterial adhesion to different structures of substrates. This pilus has a hydrophobic tip, with which it probably bonds to hydrophobic surfaces of various substrates. Therefore, the use of hydrophilic materials instead of hydrophobic plastics, especially in medical devices, can help reduce cases of hospital infections with *A. baumannii* [38].
*Enterococcus* sp.	Enterococci are Gram-positive lactic acid bacteria found in the intestinal microbiota, easily adapted to pH, temperature, and salt content conditions. Due to this versatility, its easy transmission is inevitable, even more so when bacteria such as *S. faecalis* and *S. faecium* predominate, since the ability of these enterococci to form biofilms increases their resistance to antibiotics. Enterococci have moderate resistance to aminoglycosides and intrinsic resistance to beta-lactams. When a bacterial infection is triggered (more common in patients with weakened immunity), they are capable of forming a niche of resistance genes that are easily transferred to other bacterial species, further aggravating the infection [39]. Unlike *Pseudomonas* and *Bacillus* biofilm formation, *Enterococcus* sp. biofilm formation is still difficult to understand [40,41]. In *E. faecalis,* several factors seem to be involved. Initially, the enterococci in their planktonic state adhere to the surface and, in this process, many adhesins are present, such as endocarditis and biofilm-associated pilus (Ebp), aggregating substance (Agg), enterococcal surface protein (Esp), *E. faecalis* collagen adhesin (Ace), proteases, and glycolipids [42]. Subsequently, the rhamnopolysaccharide matrix is formed, a process that has not yet been identified as to the mechanism by which enterococci form a thick or thin biofilm or why they could spread again [42]. However, it is known that the formation of microcolonies is characteristic of a mature stage of the biofilm, which is important for intestinal colonization in response to antibiotics [43,44]. Previous studies showed that the *dltABCD* operon had a strong relationship with biofilm formation and that its genetic deletion decreased biofilm formation; which allowed a better action of the antimicrobials, mainly of the antimicrobial peptides (AMPs) [45].Vancomycin-resistant *Enterococcus faecium* (VRE) is considered a priority on the WHO list, since it is also a great biofilm former and the third largest cause of nosocomial infection, especially in patients who require long-term invasive medical devices [46,47]. Several genes involved in in vitro biofilm development were reported, such as *atlA, ebpABC, esp, fsrB, luxS, spx, acm, scm, sgrA, pilA, pilB, ecbA* and *asrR,* but only some such as *atlA, ebpABC, esp, acm* and *asrR* would be directly associated with the development of a serious infection in vivo [42]. Due to controversial results of *Esp* expression, it cannot be clarified whether or not biofilm formation is mediated by enterococcal biofilm regulator B [48,49]. Although it is believed that virulence genes are related to the formation of biofilms, it was shown that it is more frequently associated with adhesion properties [47].
*Staphylococcus aureus*	Vancomycin-resistant, vancomycin-intermediate, and methicillin-resistant *Staphylococcus aureus* (VRSA, VISA, and MRSA, respectively) are known as a major cause of bacteremia, infective endocarditis, osteoarticular infections, skin and soft tissue infections, pleuropulmonary infections, and invasive-device-related infections. These bacteria can express several resistance mechanisms and virulence factors, which gives them the ability to evade the host’s natural defenses and some drugs [50]. Biofilms are generally associated with chronic and serious acute infections, and the same can be said of the biofilm of *S. aureus,* which results in a complicated treatment because many of its clinical isolates are MDR bacteria [51]. *S. aureus* has the ability to express a diverse set of virulence factors that allow it to cause a broad range of pathologies and survive the host immune system. Additionally, heterogeneous biofilms that present *S. aureus* have a high rate of plasmid horizontal transfer, which increases their antibiotic resistance [24,51]. *S. aureus* has a multilayered biofilm with a matrix composed of about 80% teichoic acids, staphylococcal proteins, and host proteins [24]. It was previously reported that GTPases would be important biofilm production markers in *S. aureus* because their inhibition enabled these bacteria to reduce the size of the biofilm and the number of mature 70S ribosomes, allowing the action of conventional antibiotics [52]. In *S. areus* biofilm-forming, tolerant, resistant, and persistent bacterial cells have been detected; the last group of bacteria is believed to be tolerant to antibiotics, causing recurrent infections and producing antibiotic-susceptible offspring when they resume growth without antibiotics [53].Previous studies recognize that the regulation of *Spx* expression would be negatively related to biofilm formation, which means that the lower the expression, the greater the maturity of the biofilm matrix and its resistance to stress [54]. It was even reported that this microorganism obtained vancomycin resistance genes (*VanA*) through gene transfer through VRE biofilms (SA + *VanA* → VRSA) [55,56]. The scarcity of reports focused on the elimination of multispecies biofilms leads us to think that we are still far from finding an ideal mechanism and drug to eliminate these perfectly organized microcolonies [57].
*Klebsiella pneumoniae*	*Klebsiella pneumoniae* (KP) is known to cause many nosocomial infections, particularly in patients using long-term medical devices. One of its virulence factors (also associated with antibiotic resistance) is biofilm formation. Growth of *K. pneumoniae* on biotic and abiotic surfaces is facilitated by type-3 fimbrial proteins (the MrkA type-3 fimbrial protein for abiotic and adhesin MrkD for biotic surfaces), which have regulated gene expression. Structures such as bacterial capsule and KP’s LPS also contribute to biofilm formation, with LPS being important in the initial adhesion to abiotic surfaces and capsule being important in the structure of a biofilm both in the early and mature stages [58].
*Mycobacterium tuberculosis*	Tuberculosis, caused by *Mycobacterium tuberculosis*, is a chronic disease in which the microorganism evades the immune system (hiding in granulomas). There are treatment failures and high reinfection rates, characteristic of infections that exhibit biofilm formation. Cellulose is an important component of *M. tuberculosis* biofilms, and since humans do not produce this compound, its presence in the lungs of infected people points to biofilm formation. Biofilms play an essential role in establishing *M. tuberculosis* infection and protection of resident bacilli from the immune system responses and consequent antimycobacterial agents. Symptoms result from the activation of the immune system when some *M. tuberculosis* bacilli invade and reside in macrophages [59].

**Table 2 pharmaceutics-14-00642-t002:** Examples of antimicrobial peptides (AMPs) and their potential antibacterial properties against infectious, MDR bacteria and some infective fungi and viruses.

AMP	Sequence	Microbial Strains	Highlights	Reference
**Antibacterial Activity**				
Cec4	GWLKKIGKKIERVGQNTRD ATIQAIGVAQQAANVAATLKGK	*A. baumannii*	Mechanism of action based on bacterial membrane rupture. It has activity against standard *A. baumannii* and MDR strains.	[75]
OM19R (MDAP-2 + Oncocin)	VDKPPYLPRPR PIRRPGGR	*E. coli, Salmonella,* and *Shigella*	Peptide with great antibacterial activity and no cytotoxicity or hemolytic properties.	[76]
ZY4	VCKRWKKWKRKWKKWCV In the sequence, disulfide bond (C-C) is formed by the Cystein.	*P. aeruginosa* and *A. baumanni*	Mechanism of action based on permeabilization of the bacterial membrane.	[77]
ARV-1502	Chex-RPDKPRPTL PRPRPPRPVR	*MDR A. baumanni*	Promising peptide when combined with standard treatment antibiotics against multi-resistant bacterial infections.	[78]
Protegrin-1	RGGRLCYCRRRFCVCVGR	*A. baumannii* (XDR andMDR strains fromsurgical wounds)	Highly active at high concentrations. Temporary effect at low concentrations.	[65]
NuriPep 1653	VRGLAPKKSLWPF GGPFKSPFN	*Pan-Drug Resistant Acinetobacter baumannii*	Activity interrupted by salt sensitivity. Thermostability at 95 °C.	[79]
Ω76 peptide	FLKAIKKFGKEFKKIGAKLK	*carbapenem- and tigecycline-resistantA. baumannii*	Mechanism of action based on the formation of an a-helical structure in bacterial membranes, causing rapid disruption, leakage, and bacterial death.	[80]
TC19	LRCMCIKWWSGKHPK	*E. coli* and *S. aureus*	High selectivity for bacterial membranes and low toxicity for human cells.	[81]
EcDBS1R6	PMKKLFKLLARIAVKIPVW	*E. coli, P. aeruginosa, K. pneumoniae, A. baumannii*	Cationic AMPs derived from a signal peptide sequence.	[82]
Iztli peptide 1 (IP-1)	KFLNRFWHWLQLKPGQPMY	*M. tuberculosis*	Mechanism of action against *M. tuberculosis* (MTB) based on the induction of autophagy in infected macrophages, thus preventing the release of MTB to new cells and directly killing the microorganism.	[83]
SET-M33 (protease resistant)	(KKIRVRLSA)4K2KβA-OH	*S. aureus* (6 strains MDR/XDR), *P. aeruginosa* (7 strains MDR/XDR)*, A. baumannii* (3 strains MDR/XDR), *E. coli* (8 strains MDR/XDR), *K. pneumoniae* (5 strains MDR/XDR)	The studies showed promising results in vitro and in vivo (5 and 2.5 mg/Kg) and showed anti-inflammatory power, decreasing the production of TNF-α, IL6, COX-2, KC, MIP-1, IP10, iNOS, NF-κB.	[84]
DP7	VQWRIRVAVIRK	*P. aeruginosa*	In vitro reduction of biofilm formation of *P. aeruginosa* from 43% to 68%.	[85]
Human β-defensin 2	HBD2/L-HBD2	*P. aeruginosa*	Inhibition of biofilm production by *A. baumannii* without reducing metabolic activity at lower concentrations.	[86]
Tilapia Piscidin 4 (TP4)	FIHHIIGGLFSAGKAI HRLIRRRRR	*P. aeruginosa, K. pneumoniae, E. coli, A. baumannii*	Cancer cells usually have anionic membranes, and many cationic AMPs such as this one have anticancer properties.	[87]
B1CTcu5	LIAGLAANFLPQILCKIARKC	*M. tuberculosis*	Mechanism of action based on the induction of morphological changes in the mycobacterial cell wall, such as cavitation and thinning of the cell wall.	[88]
CDP-B11	VRNSQSCRRNKGICV PIRCPGSMRQIGTCL GAQVKCCRRK	*A. baumannii, E. coli, P. aeruginosa, K. pneumoniae*	Mechanism of action based on inhibition of bacteria by depolarization and damaging in bacterial membranes.	[89]
EcDBS1R6	PMKKLFKLLARIAVKIPVW	*A. baumannii, E. coli, P. aeruginosa, K. pneumoniae*	Bactericidal mechanism based on induction of membrane permeabilization and loss of bacterial membrane potential.	[82]
SET-M33 protease- resistant	(KKIRVRLSA)4K2KβA-OH	*S. aureus, A. baumannii, E. coli, P. aeruginosa* and *K. pneumoniae*	Mechanism based on the strong neutralization of lipopolysaccharide (LPS) andlipoteichoic acid from bacteria. Strong anti-inflammatory effect, reducing the expression of cytokines, enzymes, and transcription factors involved in inflammatory processes.	[84]
Ctx(Ile^21^)-Ha	GWLDVAKKIGKAAFNVAKNFI	MDR *P. aeruginosa,* MDR *A. baumannii, S. aureus, E. coli* and *S. enteritidis*	Peptide originating from the frog. Promising antimicrobial activity, with physicochemical stability in different physiological conditions. Its application loaded within alginate microparticles greatly reduced hemolytic activity and even increased its bioavailability to prevent systemic infection.	[90]
NZX	GFGCNGPWSEDDIQCHNH CKSIKGYKGGYCARG GFVCKCY (Disulfide bonds at position C4–C30, C15–C37, C19–C39)	*M. tuberculosis*	Enhances the inhibition of intracellular mycobacteria in primary macrophages and preserves the ability to eliminate *M. tuberculosis* in vivo when carried into cells by nanoparticle systems such as mesoporous silica.	[91]
**Antifungal Activity**				
ARP788.14	KRWIILGLNKIVRMYSPTSI	*Candida utilis* and in vitro antifungal activity against yeast	Study based on sequence prediction of antifungal peptides using computational algorithms specialized in biological studies and, after prediction, the sequences were tested in disk diffusion and broth microdilution methods, where some promising ones were obtained.	[92]
ARP788.13	PPIPVGEIYKRWIILGLNK	in vitro antifungal activity against yeast		
Ctn[15–34] (the C-terminal fragment of Crotalicidin peptide)	KKRLKKIFKKPMVIGVTIPF	*Candida albicans* biofilms	Its mechanism of action is based on its interaction with the fungal plasma membrane followed by its disruption, in addition to preventing biofilm formation or eradicating biofilm already present. Other Crotalicidin peptide fragments are also studied for having other properties (antimicrobial, antiparasitic and antiviral) and antiproliferative (antitumor) properties.	[93]
ToAP2	FFGTLFKLGSKLIPGVMKLFSKKKER	*Candida albicans* in its planktonic form and in a biofilm colony	This peptide increases the permeability of the plasma membrane of *C. albicans* in its planktonic and biofilm form.In addition, it is effective at different stages of biofilm formation; that is, it acts on both newly formed and mature biofilms.	[94]
NDBP-5.7	ILSAIWSGIKSLF-NH2		This peptide also increases the permeability of the plasma membrane of *C. albicans* in its planktonic and biofilm form and, in addition, studies suggest that this molecule causes intracellular morphological changes in the fungus.	
MCh-AMP1	LSVKAFTGIQLRGVCGIEVKARG	*Candida albicans*	Peptide derived from the plant *Matricaria chamomilla*. Its mechanism of action is based on the interaction with the fungal membrane (the peptide is positively charged and contains hydrophobic residues, which causes it to interact with the negative components of the fungal membrane such as phosphatidylserine and phosphatidylinositol), making it more permeable. and causing loss of ions to the pathogenic cell.	[95]
KW2	KWKW-NH2	*C. albican, C. catenulate, C. intermidia, C. rugosa, C. glabrata* and *C. melibiosica*	According to the reference, the antifungal activity of these peptides increases as the peptide is extended, however, the extra amino acid residues of KW5 reduce its selectivity, despite having good antifungal activity, as does KW4. The authors also indicate that there must be a balance of cationicity and hydrophobicity for activity against *C. albicans*, including multidrug-resistant strains of C. *albicans*.	[96]
KW3	KWKWKW-NH2			
KW4	KWKWKWKW-NH2			
KW5	KWKWKWKWKW-NH2			
**Antiviral Activity**				
P9R	NGAICWGPCPTAFRQIGNCGRFRVRCCRIR	Enveloped coronaviruses (SARS-CoV-2, SARS-CoV, and MERS-CoV), influenza virus, and non-enveloped rhinovirus.	The positive charge of this peptide is essential for its antiviral activity, as it targets the inhibition of the virus–host endosomal acidification process (a key step in the life cycle of many pH-dependent viruses). The positive charge inhibits this acidification.	[97]
Piscidin-1	FFHHIFRGIVHVGKTIHRLVTG	PRV (pseudorabies virus), PEDV (porcine epidemic diarrhea virus), PRRSV (porcine reproductive and respiratory syndrome virus), TGEV (transmissible gastroenteritis virus), RV (rotavirus)	It is a polypeptide of natural origin, produced by fish.It has a potent effect on viruses such as catfish virus, frogvirus, and HIV-1. Furthermore, piscidin-1 has also been shown to have inhibitory effects on several common porcine pathogenic viruses.	[98]
Caerin 1.1	GLLSVLGSVAKHVLPHVVPVIAEHL		It is a peptide derived from a granule from the skin glands of an Australian frog. Its activity against bacteria and viruses is based on the destruction of the pathogen’s integrity by forming pores in its membrane.	
pBD-2 (Porcine β-Defensin-2)	DHYICAKKGGTCNFSPCPLFNRIEGTCYSGKAKCCIR	PRV (pseudorabies virus), PRRSV (porcine reproductive and respiratory syndrome virus)	This peptide belongs to the group of defensins, a group of cationic antibacterial peptides divided into α-, β- and θ-.The β-defensins family is mainly expressed in epithelial cells of animal skin, respiratory tract and gastrointestinal tract. Currently, more than 30 β-defensins are known in humans.	

**Table 3 pharmaceutics-14-00642-t003:** Promising AMPs against biofilm formation, potential antibacterial properties, and highlights of the promising results.

Peptide	Sequence and Properties	Antimicrobial Activity	Highlights	Reference
Myxinidin2 Myxinidin3	KIKWILKYWKWS RIRWILRYWRWS	*P. aeruginosa, S. aureus,* and *L. monocytogenes*	Effects against a wide range of bacteria, with its mechanism of action based on its ability to insert into bacterial membranes to produce an ion channel or pore that disrupts membrane function.	[106]
Colistin (colistin–imipenem and colistin–ciprofloxacin)	ALYKKLLKKLLKSAKKLG	*Pseudomonas aeruginosa, Escherichia coli* and *Klebsiella pneumoniae*	Bactericidal mechanism by a detergent-like effect. Recommended as a last choice in the treatment of infections caused by MDR Gram-negative bacteria because it rarely causes bacterial resistance.	[107]
S4(1–16)M4Ka	ALWKTLLKKVLKAAAK-NH2	*P. fluorescens*	Greater antimicrobial effect and less toxicity than its parent peptide (dermaseptin S4)	[108]
Pexiganan	GIGKFLKKAKKFGKAFVKILKK-NH2	*S. aureus, S. epidermidis, S. pyogenes, S. pneumoniae, E. coli* and *P. aeruginosa*	Weak anti-biofilm agent against structures formed on CL.	[109]
Citropin 1.1	GLFDVIKKVASVIGGL-NH2		Potent anti-biofilm agent against *S. aureus* strains.	
Temporin A:	FLPLIGRVLSGIL-NH2		Strong activity against vancomycin-resistant strains.	
Palm-KK-NH2	Palm-KK-NH2 (Palm–hexadecanoic acid residue)		Effective against most strains in the form of a biofilm. Activity potentiated when combined with standard antibiotics.	
Palm-RR-NH2	Palm-RR-NH2 (Palm–hexadecanoic acid residue)		Efficiency potentiated when combined with standard antibiotics.	
HB AMP	KKVVFWVKFK + HAp-binding heptapeptide (HBP7)	*S. mutans, L. acidophilus* and *A. viscosus*	Adsorption capacity on the dental surface.	[110]
KSLW	KKVVFWVKFK		Promising peptide for oral use as it is resistant to the gastrointestinal tract and stable in human saliva.	
TiBP1-GGG-AMP	RPRENRGRERGKGGGLKLLKKLLKLLKKL	*S. mutans, S. epidermidis,* and *E. coli.*	Bifunctional peptide capable of binding to titanium materials, enabling its use in biomaterials. Antibacterial functionality.	[100]
BA250-C10	RWRWRWK(C_10_)	*P. aeruginosa*	Great activity when used in synergism with two conventional anti-pseudomonas antibiotics to inhibit the planktonic growth of four strains of *P. aeruginosa.*	[111]
D-HB43	FAKLLAKLAKKLL	*Methicillin-resistant S. aureus strains*	High cytotoxic and hemolytic effect.	[112]
D-Ranalexin	FLGGLIKIVPAMICAVTKKC	*Methicillin-resistant S. aureus strains*	Effective in dose-dependent biofilm killing, but high cytotoxic and hemolytic effect.	
FK13-a1	WKRIVRRIKRWLR-NH2	*Methicillin-resistant S. aureus,* MDR *P. aeruginosa* and *vancomycin-resistant E. faecium*	Mechanism of action based on the induction of cytoplasmic membrane potential loss, permeabilization, and rupture.	[113]
FK13-a7	WKRWVRRWKRWLR-NH2	*Methicillin-resistant S. aureus,* MDR *P. aeruginosa* and *vancomycin-resistant E. faecium*	Mechanism of action based on the induction of cytoplasmic membrane potential loss, permeabilization, and rupture.	
KR-12-a5	KRIVKLILKWLR-NH2	*E. coli, P. aeruginosa, S. typhimurium, S. aureus, B. subtilis, S. epidermidis*	This peptide and its analogs kill microbial cells by inducing loss of cytoplasmic membrane potential, permeabilization, and disruption.	[114]
AMP2	KRRWRIWLV	*E. coli, P. aeruginosa, S. aureus, E. faecalis, S. epidermidis*	76% reduction of the biofilm area.	[115]
GH12	GLLWHLLHHLLH-NH2	*S. mutans*	Antimicrobial activity against cariogenic bacteria and its biofilms in vitro.	[116]
TP4	FIHHIIGGLFSAGKAIHRLIRRRRR	*P. aeruginosa, K. pneumoniae, S. aureus*	Peptide driven into helix shape by an LPS-like surfactant before binding to the target.	[117]
LyeTxI	IWLTALKFLGKNLGKHLALKQQLAKL	*F. nucleatum, P. gingivalis, A. actinomycetemcomitans*	Active against periodontopathic bacteria. Rapid bactericidal effect, prevention of biofilm development. Can be used in the dental field.	[118]
Esc(1–21)	*GIFSKLAGKKIKNLLISGLKG-NH2*	*P. aeruginosa*	Mechanism of action causes membrane thinning.	[119]
L12	LKKLLKKLLKKL-NH2	*P. aeruginosa, K. pneumoniae, S. aureus, E. coli*	Mechanism of action based on pore formation, inducing rapid permeabilization of bacterial membranes, inhibition of biofilm formation, disruption of drug-resistant biofilms, and suppression of LPS-induced pro-inflammatory mediators, even at low peptide concentrations.	[120]
W12	WKKWWKKWWKKW-NH2		Suppression of LPS-induced pro-inflammatory mediators, even at low peptide concentrations.	
WLBU2	RRWVRRVRRVWRRVVRVVRRWVRR	*E. faecium, S. aureus, K. pneumoniae, A. baumannii, P. aeruginosa* and *Enterobacter* species	Mechanism of action based on preventing bacterial adhesion and interfering with gene expression.	[121,122]
LL37	LLGDFFRKSKEKIGKEFKRIVQRIKDFLRNLVPRTES	*E. faecium, S. aureus, K. pneumoniae, A. baumannii, P. aeruginosa* and *Enterobacter* species	One of the most important human AMPs that play roles in the defense against local and systemic infections. Bactericidal mechanism against Gram-positive and Gram-negative bacteria based on phospholipid-dependent bacterial membrane disruption.	[121,123]
SAAP-148	LKRVWKRVFKLLKRYWRQLKKPVR	*E. faecium, S. aureus, K. pneumoniae, A. baumannii, P. aeruginosa* and *Enterobacter* species	Promising peptide fights difficult-to-treat infections due to its broad antimicrobial activity against MDR, biofilm, and persistent bacteria.	[124]
WAM-1	KRGFGKKLRKRLKKFRNSIKKRLKNFNVVIPIPLPG	*A. baumannii*	This peptide originates from LL37 AMPs and is more effective in inhibiting biofilm dispersion than its parent peptide.	[125]
H4	KFKKLFKKLSPVIGKEFKRIVERIKRFLR	*S. aureus, S. epidermidis, S. pneumoniae, E. coli, E. faecium, K. pneumoniae,* and *P. aeruginosa*	Insignificant rates of toxicity to eukaryotic cells.	[126]
RWRWRWA-(Bpa)	RWRWRWA-(4-benzophenylalanine)	*P. aeruginosa*	It targets the bacterial lipid membrane, but there is no specific receptor. It only affects a range of cellular processes.	[127]
Pse-T2	LNALKKVFQKIHEAIKLI-NH2	*P. aeruginosa, S. aureus, E. coli*	Mechanism of action based on the ability to disrupt the outer and inner membrane of Gram-negative bacteria and to bind DNA.	[128]
Magainin 2	GIGKFLHSAKKFGKAFVGEIMNS-NH2	*A. baumannii strains*	Strong antibacterial activity against *A. baumannii,* including MDR strains. Non-toxic to mammalian cells.	[129]
Magainin I	GIGKFLHSAGKFGKAFVGEIMKS	*E. coli strains*	Demands more energy metabolism, translational processes, and bacterial defense in *E. coli* strains when present.	[130]
TC19	LRCMCIKWWSGKHPK	*B. subtilis strains*	Promising peptide against Gram-positive bacteria, as its activity on the membrane interferes with several essential cellular processes, leading to bacterial death.	[131]
TC84	LRAMCIKWWSGKHPK		Promising peptide against Gram-positive bacteria, as its activity on the membrane interferes with several essential cellular processes, leading to bacterial death.	
BP2	GKWKLFKKAFKKFLKILAC	*B. subtilis strains*	Promising peptide against Gram-positive bacteria, as its activity by perturbation of the membrane interferes with several essential cellular processes, leading to bacterial death.	[132]
Nisin A	MSTKDFNLDLVSVSKKDSGASPRITSISLCTPGCKTGALMGCNMKTATCHCSIHVSK	*B. subtilis spores*	Application as an adjuvant to antibiotic peptides in providing a bactericidal coating for the spores.	[131,133]

## Data Availability

Not applicable.

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
