# Peer review of "Antimicrobial Peptides as an Alternative for the Eradication of Bacterial Biofilms of Multi-Drug Resistant Bacteria"

_pharmaceutics, 2022, doi:10.3390/pharmaceutics14030642_

Round 1

Reviewer 1 Report

The article titled “Antimicrobial peptides as an alternative for the eradication of bacterial biofilms of multi-drug resistant bacteria” may be a useful contribution to the journal; The authors have categorized all the sections well. The description of the content is excellent, but the figures and table must be revised.

  1. In figure 1, the phrase written in the figure is not so clear and a little blurry to understand. For readers, the quality of the pictures and the clarity of the text must be improved.
  2. In table 1, the highlights column for reference number 82 must be elaborated.
  3. Please abbreviate appropriately, it must be written AMPs instead of AMP in table 1. Please check throughout the article and revise it accordingly.
  4. When the author gives a bacterial species names as a caption, it must be written in either “italics” or “full caps” – line number 233, 248, 261, 294, 323, & 332.
  5. A paragraph that explains the detail of multi-drug resistant bacteria should be inserted in the Introduction section.
  6. The authors should highlight the novelty of their work, as compared to others in the field.
  7. In Table 1, if the author studied only the antibacterial activity of AMPs then please revise the column name into “Antibacterial activity” instead of “Antimicrobial activity”
  8. In Table 2, reference number 121 the author discussed the antifungal activity of AMPs, so the figure caption must be changed and please use the term “Antimicrobial activity” instead of “Antibacterial activity”.
  9. The author mainly studied Antimicrobial peptides (AMPs) and their alternative use in the eradication of bacterial biofilms. In line number 343-345 - the author mentioned “AMPs are biomolecules formed by amino acids that vary in length, usually composed of 12–50 amino acids [64]. They are known to have great antifungal, antiviral, and antibacterial properties”. It is expected to see more of their antimicrobial activities instead of antibacterial activity alone. Please include some of the antifungal, and antiviral activities of AMPs in Table 1 and Table 2.
  10. Please discuss the future perspectives of the work in Section 3.

Author Response

REVIEWER 1

The article titled “Antimicrobial peptides as an alternative for the eradication of bacterial biofilms of multi-drug resistant bacteria” may be a useful contribution to the journal; The authors have categorized all the sections well. The description of the content is excellent, but the figures and table must be revised.

In figure 1, the phrase written in the figure is not so clear and a little blurry to understand. For readers, the quality of the pictures and the clarity of the text must be improved.

We appreciate your comment, the figure was improved.

In table 1, the highlights column for reference number 82 must be elaborated.

Thank you for the comment, we explained the highlight in the reference.

Please abbreviate appropriately, it must be written AMPs instead of AMP in table 1. Please check throughout the article and revise it accordingly.

Thank you for the comment, Abbreviations have been carefully checked.

When the author gives a bacterial species names as a caption, it must be written in either “italics” or “full caps” – line number 233, 248, 261, 294, 323, & 332.

The text was modified and according to the second reviewer, it was transferred to a new table.

A paragraph that explains the detail of multi-drug resistant bacteria should be inserted in the Introduction section.

The following statement was added:

Bacterial resistance can be innate or acquired since many bacteria have resistance genes that are expressed only when they feel threatened; and they can easily transfer these genes when found in microecosystems to other species for community survival.

The authors should highlight the novelty of their work, as compared to others in the field.

The following statement was added:

To our knowledge, this is the first article reporting the importance of AMPs against bio-film-forming bacteria on the WHO critical and high priority list.

In Table 1, if the author studied only the antibacterial activity of AMPs then please revise the column name into “Antibacterial activity” instead of “Antimicrobial activity”

We appreciate your comment, we have expanded table 1 (new table 2).

In Table 2, reference number 121 the author discussed the antifungal activity of AMPs, so the figure caption must be changed and please use the term “Antimicrobial activity” instead of “Antibacterial activity”.

We appreciate your comment, we have expanded table 1 (new table 2) with AMPs against fungi, bacteria and viruses.

The author mainly studied Antimicrobial peptides (AMPs) and their alternative use in the eradication of bacterial biofilms. In line number 343-345 - the author mentioned “AMPs are biomolecules formed by amino acids that vary in length, usually composed of 12–50 amino acids [64]. They are known to have great antifungal, antiviral, and antibacterial properties”. It is expected to see more of their antimicrobial activities instead of antibacterial activity alone. Please include some of the antifungal, and antiviral activities of AMPs in Table 1 and Table 2.

We appreciate your comment, we have expanded table 1 (new table 2) and but in the table 2 (new table 3) we eliminate anti-Candida sp. activity, since the focus of the review is against biofilm-forming bacteria on the WHO priority list.

Please discuss the future perspectives of the work in Section 3.

The section has been briefly improved and expanded.

Reviewer 2 Report

The submitted manuscript “Antimicrobial peptides as an alternative for the eradication of bacterial biofilms of multi-drug resistant bacteria” by de Pontes et al summarised the antimicrobial peptide as new biomacromolecules for dangerous and highly infectious bacteria. However, the main content of the manuscript are focusing on AMR development, AMR mechanism and basic information on biofilm formation. There is only a short description of the AMPs in the manuscript, which is not reflecting the title suggested in this manuscript. The authors strongly recommend addressing this part.

There are a few additional major comments,

  1. Some abbreviations were not consistent, for example, antimicrobial resistance or AMR, AMPs or antimicrobial peptides were used a couple of times in the text.
  2. Line 94-98, this is repeated with the AMR mechanism section 1.2.
  3. Line 193-195, the ref is missing to support this statement.
  4. Line 223, they haven’t introduced AMP yet in the whole section, why they put AMP here. The flow of the manuscript should be carefully checked.
  5. Line 232, it will be better to summarise a table of the main biofilm-forming bacteria.
  6. Line 347, for AMPs introduction, the recent excellent summary on AMPs for antibacterial should be discussed (Lancet Infect Dis 2020; 20: e216–30, https://doi.org/10.1016/S1473-3099(20)30327-3, Chem. Soc. Rev., 2021,50, 4932-4973 https://doi.org/10.1039/D0CS01026J).
  7. Line 360, recent work on antibiofilm AMP works should be discussed in the manuscript (Nat Rev Microbiol (2021). https://doi.org/10.1038/s41579-021-00585-w, Chem. Sci. 2022, 13, 2226-2237. DOI https://doi.org/10.1039/D1SC05662J )
  8. Line 366, according to Yasir…, a ref is missing.

Author Response

REVIEWER 2

The submitted manuscript “Antimicrobial peptides as an alternative for the eradication of bacterial biofilms of multi-drug resistant bacteria” by de Pontes et al summarised the antimicrobial peptide as new biomacromolecules for dangerous and highly infectious bacteria. However, the main content of the manuscript are focusing on AMR development, AMR mechanism and basic information on biofilm formation. There is only a short description of the AMPs in the manuscript, which is not reflecting the title suggested in this manuscript. The authors strongly recommend addressing this part.

There are a few additional major comments,

Some abbreviations were not consistent, for example, antimicrobial resistance or AMR, AMPs or antimicrobial peptides were used a couple of times in the text.

We appreciate your comment and we modify the abbreviations quoting for the first time for each structural part of the manuscript i.e., abstract, body, tables, figures.

Line 94-98, this is repeated with the AMR mechanism section 1.2.

We appreciate your comment, we have eliminated that paragraph.

Line 193-195, the ref is missing to support this statement.

Thank you for the comment, we added a reference.

Line 223, they haven’t introduced AMP yet in the whole section, why they put AMP here. The flow of the manuscript should be carefully checked.

The paragraph has been removed and relocated to the AMP and Applications section.

Line 232, it will be better to summarise a table of the main biofilm-forming bacteria.

As suggested, we include all the bacteria studied in the manuscript in a table.

Line 347, for AMPs introduction, the recent excellent summary on AMPs for antibacterial should be discussed (Lancet Infect Dis 2020; 20: e216–30, https://doi.org/10.1016/S1473-3099(20)30327-3, Chem. Soc. Rev., 2021,50, 4932-4973 https://doi.org/10.1039/D0CS01026J).

Thanks for your suggestion, information has been expanded to improve the section on antimicrobial peptides.

Line 360, recent work on antibiofilm AMP works should be discussed in the manuscript (Nat Rev Microbiol (2021). https://doi.org/10.1038/s41579-021-00585-w, Chem. Sci. 2022, 13, 2226-2237. DOI https://doi.org/10.1039/D1SC05662J )

We appreciate your comment, we have expanded the discussion before Table 3.

Line 366, according to Yasir…, a ref is missing.

We apologize, the reference was added.

Round 2

Reviewer 2 Report

The authors have addressed my comments.